# Handshape-Aware Sign Language Recognition:
# Extended Datasets and Exploration of Handshape-Inclusive Method

**Xuan Zhang**
Johns Hopkins University
xuanzhang@jhu.edu

**Kevin Duh**
Johns Hopkins University
kevinduh@cs.jhu.edu

## Abstract

The majority of existing work on sign language recognition encodes signed videos without explicitly acknowledging the phonological attributes of signs. Given that handshape is a vital parameter in sign languages, we explore the potential of handshape-aware sign language recognition. We augment the PHOENIX14T dataset with gloss-level handshape labels, resulting in the new PHOENIX14T-HS dataset. Two unique methods are proposed for handshape-inclusive sign language recognition: a single-encoder network and a dual-encoder network, complemented by a training strategy that simultaneously optimizes both the CTC loss and frame-level cross-entropy loss. The proposed methodology consistently outperforms the baseline performance. The dataset and code can be accessed at https://github.com/Este1le/slr_handshape.git.

## 1 Introduction

Sign languages are primarily the languages of Deaf people. They are the center of the Deaf culture and the daily lives of the Deaf community. [1] In the U.S., estimates suggest that between 500,000 to two million people communicate using American Sign Language (ASL), making it the fifth most-used minority language in the country after Spanish, Italian, German, and French (Lane et al., 1996). Natural sign languages, which develop independently and possess unique grammatical structures distinct from surrounding spoken languages, are just as crucial to include in the field of natural language processing (NLP) as any other language, as Yin et al. (2021) advocates.

One direction in sign language processing (SLP) is sign language recognition (SLR), a task of recognizing and translating signs into glosses, the writ-

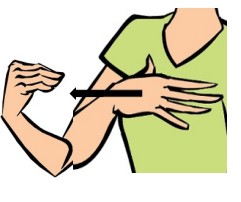 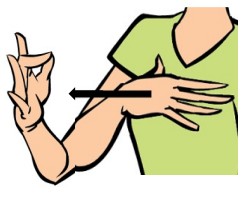

**WHITE**        **LIKE**

Figure 1: An example of a handshape minimal pair in ASL.[2] Both signs start from an identical handshape but end with a distinct one. In practical scenarios, when a signer signs rapidly, the terminal handshape of **LIKE** may closely resemble that of **WHITE**, leading to potential difficulties in differentiation.

ten representations of signs typically denoted by spoken language words. Among the array of SLR products, sign gloves, and wearable devices using sensors to track hand movements, are widespread. However, these devices have faced criticism from the Deaf community, primarily due to the social stigma associated with wearing them. This feedback has motivated us to explore video-based SLR, an alternative approach that utilizes cameras to record signs and feed them into the system as video inputs. By doing so, we aspire to foster a more inclusive society, preserving the valuable sign languages serving as the heart of the Deaf culture, and thereby facilitating improved communications between the Deaf and hearing communities.

Signs can be defined by five parameters: handshape, orientation, location, movement, and non-manual markers such as facial expressions. Signs that differ in only one of these parameters can form minimal pairs. An example of a handshape minimal pair in ASL is illustrated in Figure 1. As reported by Fahey and Hilger (2022), among all parameters, handshape minimal pairs are identified with the lowest accuracy – only 20%, compared to palm orientation (40%), location (47%), and movement (87%). This indicates the complexity in-

---

[1] Deaf sociolinguist Barbara Kannapell: "It is our language in every sense of the word. We create it, we keep it alive, and it keeps us and our traditions alive." And further, "To reject ASL is to reject the Deaf person."

[2] Images clipped from https://babysignlanguage.com/.

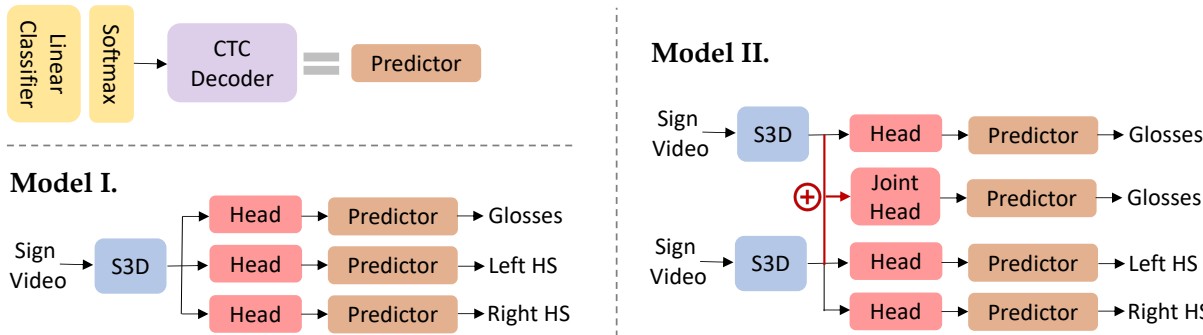

Figure 2: We propose two handshape-inclusive SLR network variants. **Model I** employs a single video encoder, while **Model II** implements both a gloss encoder and a handshape (HS) encoder, applying a joint head to the concatenated representations produced by the two encoders.

volved in distinguishing handshapes, underscoring their importance in correctly interpreting signs.

The majority of existing research on SLR does not incorporate phonological features such as handshapes into their system designs, with only a few exceptions (Koller et al., 2016; Cihan Camgoz et al., 2017; Koller et al., 2019). Typically, signs are interpreted as a cohesive whole, meaning that an SLR model is expected to correctly recognize all five parameters simultaneously to accurately identify a sign. This constitutes a major distinction between spoken and sign languages – the former is linear, while the latter incorporates both linearity and simultaneity (Hill et al., 2018). This uniqueness introduces considerable challenges to SLR tasks.

The limited interest in integrating handshapes into SLR systems can be attributed largely to the absence of handshape annotations in existing Continuous SLR (CSLR[3]) datasets. In response to this, we have extended one of the most widely used SLR datasets, PHOENIX14T, with handshape annotations, sourced from online dictionaries and manual labeling, thus creating the PHOENIX14T-HS dataset. Our hope is that this will facilitate more research into handshape-aware SLR.

Moreover, we introduce two handshape-inclusive[4] SLR networks (Figure 2), designed with either single or dual-encoder architectures. These proposed models extend the basic SLR network, which doesn't include handshape information in gloss prediction. Thus, any existing SLR can adopt

our approach, underscoring the adaptability of our methods.

We set a benchmark on the PHOENIX14T-HS dataset with the proposed methods. Our models outperform previous state-of-the-art (SOTA) single-modality SLR networks, which utilize only RGB videos as input and were trained on PHOENIX14T.

## 2 Related Work

### 2.1 SLR

In recent developments of CSLR, a predominant methodology has emerged that employs a hybrid model. The model is usually composed of three essential components: a visual encoder, which extracts the spatial features from each frame of the sign video; a sequence encoder, responsible for learning the temporal information; and an alignment module which monotonically aligns frames to glosses. The visual encoder component could be built with various architectures, including 2D-CNNs (Koller et al., 2019; Cheng et al., 2020; Min et al., 2021), 3D-CNNs (Chen et al., 2022a,b), or 2D-CNNs followed by 1D-CNNs (Papastratis et al., 2020; Pu et al., 2020; Zhou et al., 2021a,b). The sequence encoder can be implemented using LSTMs (Cui et al., 2019; Pu et al., 2020; Zhou et al., 2021b), Transformer encoders (Niu and Mak, 2020; Camgoz et al., 2020; Zuo and Mak, 2022; Chen et al., 2022b), or 1D-CNNs (Cheng et al., 2020). In terms of the alignment module, research attention has been redirected from HMM (Koller et al., 2017, 2019) to connectionist temporal classification (CTC) (Hao et al., 2021; Zhou et al., 2021b; Zuo and Mak, 2022; Chen et al., 2022b).

Various approaches have been proposed to improve SLR system performance.

---

[3]CSLR refers to the recognition of sign language at the sentence level, as opposed to Isolated SLR, which operates at the word level. Our work focuses on CSLR due to its broader practical application and a higher level of complexity.

[4]We use *handshape-aware* to denote SLR that incorporates handshape information during training, while *handshape-inclusive* pertains to the deliberate inclusion of handshape predictions within SLR.

**Multi-stream network** The multi-stream networks use multiple parallel encoders to extract features from distinct input streams. In addition to the RGB stream, Cui et al. (2019) incorporate an optical flow stream, while Zhou et al. (2021b); Chen et al. (2022b) use key points. Koller et al. (2019) and Papadimitriou and Potamianos (2020) introduce two extract encoders for hand and mouth encoding, directing the system's focus towards critical image areas.

**Cross-entropy loss** Training objectives beyond CTC loss can also be employed. Cheng et al. (2020); Hao et al. (2021) train their models to also minimize the frame-level cross-entropy loss, with frame-level labels derived from the CTC decoder's most probable alignment.

## 2.2 Handshape-inclusive Datasets

Currently, datasets frequently employed for the continuous SLR task, such as RWTH-PHOENIX-Weather 2014T (Camgoz et al., 2018) and CSL Daily (Zhou et al., 2021a) generally lack handshape annotations, except for RWTH-PHOENIX-Weather 2014 (Koller et al., 2015), which is extended by Forster et al. (2014) with handshape and orientation labels. The annotating process involved initially labeling the orientations frame-by-frame, followed by clustering within each orientation, and then manually assigning a handshape label to each cluster. Additionally, a subset of 2k signs is annotated using the SignWriting (Sutton and DAC, 2000) annotation system. To facilitate handshape recognition, Koller et al. (2016) introduced the 1-Million-Hands dataset, comprising 1 million cropped hand images from sign videos, each labeled with a handshape. The dataset consists of two vocabulary-level datasets in Danish and New Zealand sign language, where handshapes are provided in the lexicon, and a continuous SLR dataset, PHOENIX14, annotated with SignWriting. It also includes 3k manually labeled handshape test images.

## 2.3 Handshape-aware SLR

Research on leveraging handshape labels to support SLR has been relatively scarce. Koller et al. (2016) applied the statistical modelling from Koller et al. (2015) and incorporated a stacked fusion with features from the 1-Million-Hands model and full frames. While Cihan Camgoz et al. (2017) and Koller et al. (2019) utilized a multi-stream sys-tem, where two separate streams are built to predict handshapes and glosses, respectively. These two streams are then merged and trained for gloss recognition. The aforementioned studies are all carried out on the PHOENIX14 dataset, made possible by the efforts of Forster et al. (2014), which extended the dataset with handshape labels. Our work instead focuses on the PHOENIX14T dataset.

## 3 Datasets

We have enriched the SLR dataset PHOENIX14T by incorporating handshape labels derived from the SignWriting dictionary and manual labeling. In the subsequent sections, we initially present the original PHOENIX14T dataset (3.1) and the Sign-Writing dictionary (3.2), followed by a detailed description of the updated PHOENIX14T dataset (PHOENIX14T-HS), now featuring handshape labels (3.3).

## 3.1 PHOENIX14T

PHOENIX14T (Camgoz et al., 2018) is one of the few predominantly utilized datasets for SLR tasks nowadays. This dataset consists of German sign language (DGS) aired by the German public TV station PHOENIX in the context of weather forecasts. The corpus comprises DGS videos from 9 different signers, glosses annotated by deaf experts, and translations into spoken German language. Key statistics of the dataset are detailed in Table.1.

PHOENIX14T (Camgoz et al., 2018), an extension of PHOENIX14 (Koller et al., 2015), features redefined sentence segmentations and a slightly reduced vocabulary compared to its predecessor. Despite Forster et al. (2014) having expanded PHOENIX14 with handshape labels, their extended dataset is not publicly accessible and only includes labels for the right hand. In contrast, our annotated data will be released publicly, encompassing handshapes for both hands.

## 3.2 SignWriting

The SignWriting dictionary (Sutton and DAC, 2000; Koller et al., 2013) publicly accessible, user-edited sign language dataset, encompassing more than 80 distinct sign languages. Adhering to the International SignWriting Alphabet, which prescribes a standard set of icon bases, users represent signs via abstract illustrations of handshapes, facial expressions, orientations, and movements. These depictions can be encoded into XML format and

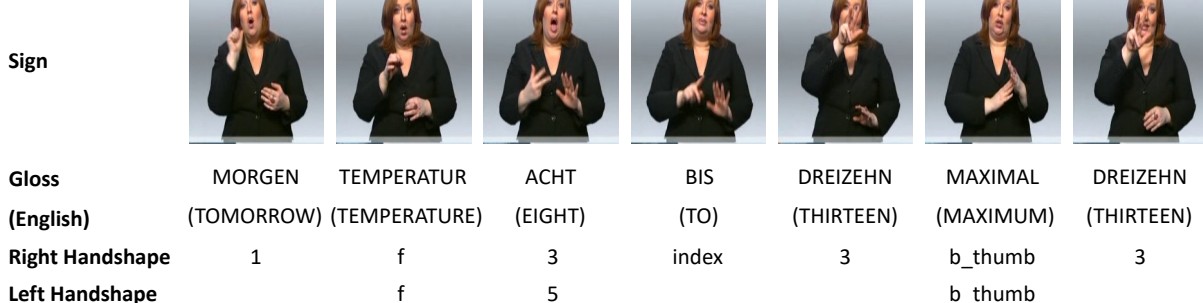

| | | | | | | | |
|---|---|---|---|---|---|---|---|
| **Sign** | | | | | | | |
| **Gloss** | MORGEN | TEMPERATUR | ACHT | BIS | DREIZEHN | MAXIMAL | DREIZEHN |
| **(English)** | (TOMORROW) | (TEMPERATURE) | (EIGHT) | (TO) | (THIRTEEN) | (MAXIMUM) | (THIRTEEN) |
| **Right Handshape** | 1 | f | 3 | index | 3 | b_thumb | 3 |
| **Left Handshape** | | f | 5 | | | b_thumb | |

Figure 3: One sample from the PHOENIX14T-HS, where handshape labels have been appended for both hands atop the PHOENIX14T dataset. When loaded into a Python environment, this sample appears as a Python dictionary: *{name: train/01April_2010_Thursday_heute-6703, signer: Signer04, gloss: MORGEN TEMPERATUR ACHT BIS DREIZEHN MAXIMAL DREIZEHN, handshape-right: [[1], [f], [3], [index], [3], [b_thumb], [3]], handshape-left: [[], [f], [5], [], [], [b_thumb], []]}.* [5]

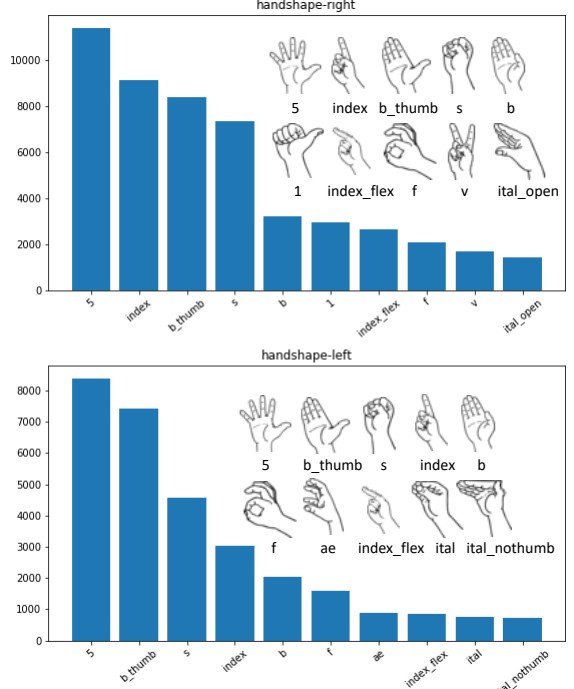

Figure 4: Top 10 most frequent handshapes for each hand in PHOENIX14T-HS.

converted into textual descriptions. We utilized the SignWriting parser[6] provided by Koller et al. (2013) to extract handshapes for both hands from the original SignWriting dictionary.

### 3.3 Handshape-extended PHOENIX14T (PHOENIX14T-HS)

There are 17,947 entries for DGS in the SignWriting dictionary. However, 314 signs/glosses in the PHOENIX14T dataset are either not included or lack handshape annotations in the dictionary (Table.1). This implies that 4,366 of the 7,096 samples in the train set contain signs devoid of handshape labels. We thus manually labeled these 314 signs.

This results in the following annotation steps:

1. Look up the SignWriting dictionary.
2. Manually label handshapes for signs not present in SignWriting.

The author, who has a competent understanding of ASL and Sign Language Linguistics, yet lacks formal training in DGS, annotated by simultaneously watching the corresponding sign video to ensure alignment. The task proves particularly demanding when consecutive gaps–signs missing handshapes–emerge. To delineate the boundaries of these signs, the author resorted to online DGS dictionaries (not SignWriting). The entire manual annotation process took around 30 hours.

Our method contrasts with that of Koller et al. (2016), which applied frame-level handshape annotations. We have instead adopted gloss-level handshape annotations. While the frame-level approach is more detailed, Koller et al. (2016) reported a significant number of blurred frames, making the task of frame-by-frame labeling both challenging and time-intensive. Moreover, given that sign language recognition is essentially a gloss-level recognition task, our aim is to maintain consistency in granularity when integrating handshape recognition as an additional task within the framework.

---

[6]https://github.com/huerlima/signwriting-parser

[6]Note that one-handed signs do not have handshape labels for the left hand, which is the non-dominant hand for this signer.

| PHOENIX14T | Train | Dev | Test |
|---|---|---|---|
| #samples | 7,096 | 519 | 642 |
| vocab. | 1,066 | 393 | 411 |
| avg. gloss len. | 7.80 | 7.23 | 6.65 |
| vocab. not in SW | 299 | 70 | 71 |
| vocab. missing HS in SW | 15 | 9 | 7 |
| #samples w/ missing HS | 4,366 | 304 | 362 |

Table 1: Statistics of PHOENIX14T. In the train set, 299 signs are absent from the SignWriting (SW), and 15 signs lack handshape (HS) annotations in SW, which indicates that 4,366 samples include signs without handshape annotations. We thus manually annotated the combined total of 314 (299+15) signs.

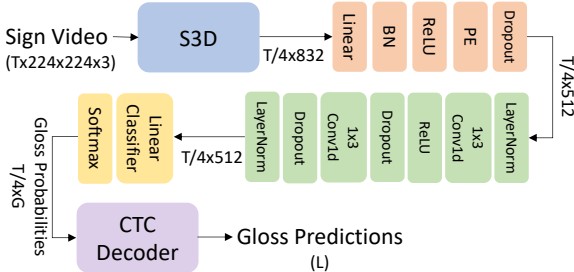

Figure 5: Architecture of the vanilla SLR network. The head network is composed of orange and green blocks. $T$ is the number of frames in the sign video. $G$ is the size of the gloss vocabulary. $L$ is the length of the predicted gloss sequence. RGB frames are resized to 224x224 during preprocessing.

We adopt the Danish taxonomy for handshape labels as in (Koller et al., 2016), which includes 60 unique handshapes. The application results in the PHOENIX14T-HS dataset, from where an example is shown in Figure 3. Given that all 9 signers in PHOENIX14T are right-hand dominant, it is appropriate to employ the default annotations from SignWriting without necessitating side-switching.

Figure 4 presents the frequency distribution of the top 10 most prevalent handshapes for each hand in the PHOENIX14T dataset. It is important to note that a single sign may comprise multiple handshapes. In fact, 13.5% of the signs in the dataset incorporate more than one handshape for the right hand, whereas the left hand employs more than one handshape in 5% of the signs.

**Limitations** We would like to note certain limitations in the proposed PHOENIX14T-HS dataset. While approximately one-third of the signs were manually labeled with handshapes, the remaining two-thirds were labeled using the user-generated SignWriting dictionary. As a result, these handshape labels may contain noise and should not be seen as curated. When dealing with sign variants, i.e. multiple entries for a single sign, our selection process was random and thus may not necessarily correspond with the sign video.

Moreover, individual signers possess unique signing preferences, leading them to opt for different sign variants. Furthermore, the signers might deviate from the dictionary-form signs, resulting in discrepancies between the real-world usage and the standardization form. In terms of our labeling process, we omitted handshape labels during the initial and final moments of each video, when the signers' hands are in a resting position. Finally,

we did not account for co-articulation, the transition phase between two consecutive signs, in our handshape labeling.

## 4 Methods

The task of SLR can be defined as follows. Given an input sign video $\mathcal{V} = (v_1, ..., v_T)$ with $T$ frames, the goal is to learn a network to predict a sequence of glosses $\mathcal{G} = (g_1, ..., g_L)$ with $L$ words, monotonically aligned to $T$ frames, where $T \geq L$.

In this section, we start by describing the vanilla SLR network, where handshapes are not provided or learned during training in Section 4.1. We then introduce the two handshape-inclusive network architectures employed in this study in Section 4.2. Specifically, these networks are designed to predict sign glosses and handshapes concurrently. Finally, we elaborate on our chosen training and pretraining strategy in Section 4.3 and 4.4.

### 4.1 Vanilla SLR networks

The architecture of the vanilla SLR network is illustrated in Figure 5. Similar to Chen et al. (2022a) and Chen et al. (2022b), we use an S3D (Xie et al., 2018) as the video encoder, followed by a head network, where only the first four blocks of S3D are included to extract dense temporal representations. Then, a gloss classification layer and a CTC decoder are attached to generate sequences of gloss predictions.

### 4.2 Handshape-inclusive SLR networks

Figure 2 depicts our proposal of two handshape-inclusive SLR network variants, which are expansions upon the vanilla network. Both variants explicitly utilize handshape information by training

the model to predict glosses and handshapes concurrently. The key distinction between the two variants lies in the employment of either a single encoder or dual encoders.

**Model I.** In comparison to the vanilla network, this model forwards the S3D feature to two additional heads, each tasked with predicting the handshapes for the left and right hand respectively. The loss for this model is computed as follows:

$$\mathcal{L}_{Model_I} = \mathcal{L}_{CTC}^G + \lambda^L \mathcal{L}_{CTC}^L + \lambda^R \mathcal{L}_{CTC}^R, \quad (1)$$

where $\mathcal{L}_{CTC}^G$ represents the CTC loss of the gloss predictor. $\mathcal{L}_{CTC}^L$ and $\mathcal{L}_{CTC}^G$ denote the CTC losses for the left and right handshape predictors, weighted by $\lambda^L$ and $\lambda^R$.

**Model II.** This model employs dual encoders, each dedicated to encoding the representations for glosses and handshapes independently. While both encoders receive the same input (sign videos) and share the same architecture, they are trained with different target labels (gloss vs. handshape). We also incorporate a joint head, which combines the visual representation learned by both encoders to generate gloss predictions. The architecture of this joint head mirrors that of the gloss head and the handshape head. Therefore, the loss for this model is computed as follows:

$$\mathcal{L}_{Model_{II}} = \mathcal{L}_{CTC}^G + \mathcal{L}_{CTC}^J + \lambda^L \mathcal{L}_{CTC}^L + \lambda^R \mathcal{L}_{CTC}^R, \quad (2)$$

where $\mathcal{L}_{CTC}^J$ denotes the CTC loss of the joint gloss predictor.

For this model, we also adopt a **late ensemble strategy**. This involves averaging the gloss probabilities predicted by both the gloss head and the joint head. The averaged probabilities are then fed into a CTC decoder, producing the gloss sequence.

### 4.3 Training strategy

The CTC loss is computed by taking the negative logarithm of the probability of the correct path, which corresponds to the true transcription. It is a relatively coarse-grained metric because it operates at the gloss level, not requiring temporal boundaries of glosses. Given that handshape prediction could potentially operate on a frame level, it stands to reason for us to compute the loss at this level as well. However, as the PHOENIX14T-HS dataset does not provide temporal segmentations, we opt to estimate these with gloss probabilities[7] generated

---
[7]For Model II, they are the averaged probabilities.

by our models. First, we extract the best path for glosses from a CTC decoder and fill in the blanks with neighboring glosses. After this, if a particular gloss has only one associated handshape, we assign that handshape to all frames within the extent of that gloss. If there is more than one handshape, we gather the handshape probabilities produced by the handshape classifiers within that segment and feed them into a CTC decoder to determine the optimal handshape labels for the frames within that gloss's range[8]. Finally, we calculate the cross-entropy loss between the pseudo-labels and the handshape probabilities. This enables more fine-grained frame-level supervision.

The loss function then becomes:

$$\mathcal{L}_{SLR} = \mathcal{L}_{Model_{I|II}} + \lambda_{CE}^L \mathcal{L}_{CE}^L + \lambda_{CE}^R \mathcal{L}_{CE}^R, \quad (3)$$

where $\mathcal{L}_{CE}^L$ and $\mathcal{L}_{CE}^R$ are cross-entropy loss for left and right hand weighted by $\lambda_{CE}^L$ and $\lambda_{CE}^R$ respectively.

### 4.4 Pretraining

Given that our target dataset is relatively sparse, it's crucial to pretrain the model to ensure a solid initialization. We first pretrain the S3D encoder on the action recognition dataset, Kinetics-400 (Kay et al., 2017), consisting of 3 million video clips. Following this, we further pretrain on a word-level ASL dataset, WLASL (Li et al., 2020), which includes 21 thousand videos.

## 5 Experiments

In this section, we present the performance of our top-performing model (Section 5.1) and further conduct ablation study (Section 5.2) to analyze the crucial components of our implementations.

### 5.1 Best model

Our highest-performing system utilizes the dual-encoder architecture of **Model II**. After initial pretraining on Kinetics-400 and WLASL datasets, we freeze the parameters of the first three blocks of the S3D. For the hyperparameters, we set $\lambda^L$ and $\lambda^R$ to 1, while $\lambda_{CE}^L$ and $\lambda_{CE}^R$ are set to 0.05. The initial learning rate is 0.001. Adam is used as the optimizer.

A comparison of our premier system (**HS-SLR**) with leading SLR methods on the PHOENIX14T

---
[8]For the left hand, when the sign does not have corresponding handshape, we label it with the special token *<pad>*.

| Method | Dev | Test |
|---|---|---|
| CNN-LSTM (Koller et al., 2019)* | 22.1 | 24.1 |
| SFL (Niu and Mak, 2020) | 25.1 | 26.1 |
| FCN (Cheng et al., 2020) | 23.3 | 25.1 |
| Joint-SLRT (Camgoz et al., 2020) | 24.6 | 24.5 |
| CMA (Papastratis et al., 2020)* | 23.9 | 24.0 |
| SignBT (Zhou et al., 2021a) | 22.7 | 23.9 |
| MMTLB (Chen et al., 2022a) | 21.9 | 22.5 |
| SMKD (Hao et al., 2021) | 20.8 | 22.4 |
| **HS-SLR(ours)** | **20.3** | **21.8** |
| STMC-R (Zhou et al., 2021b)* | 19.6 | 21.0 |
| $C^2$SLR (Zuo and Mak, 2022)* | 20.5 | 20.4 |
| TwoStream (Chen et al., 2022b)* | 17.7 | 19.3 |

Table 2: Comparison with previous work on SLR on PHOENIX14T evaluated by WER. The previous best results are underlined. Methods marked with * denote approaches that utilize multiple modalities besides RGB videos, such as human body key points and optical flow. Notably, our best model (**Model II**) achieves the lowest WER among single-modality models.

dataset is shown in Table 2. While we do not surpass the existing records, our system ranks as the top performer among all single-modality models. It is worth noting that the extension of multi-modality models into handshape-inclusive models, such as introducing handshape heads or an additional handshape encoder to the original networks, could potentially enhance the SOTA performance further.[9]

## 5.2 Ablation study

### 5.2.1 Model variants

In our analysis, we contrast our suggested model variants, **Model I** and **Model II** (discussed in Section 4.2), with the **Vanilla** SLR network (described in Section 4.1). Additionally, we compare models that feature solely a right handshape head against those equipped with two heads, one for each hand. An extended variant, **Model II+**, which adds two handshape heads to the gloss encoder, is also considered in our experimentation.

As demonstrated in Table 3, **Model II** outperforms **Model I** and **Model II+**. The performance differs marginally between models with handshape heads for both heads versus those with a single right-hand head.

---

[9]Due to computational resource constraints, we are currently unable to fit such models into our GPU devices. Future work may explore the expansion of multi-modality models to include handshape-inclusive models.

| Model | Hands | Dev | Test |
|---|---|---|---|
| **Vanilla** | - | 23.54 | 23.69 |
| **Model I*** | right | 23.22 | 22.94 |
| **Model I** | right + left | 22.52 | 23.43 |
| **Model II*** | right | 21.06 | 22.56 |
| **Model II** | right + left | **21.03** | **22.07** |
| **Model II+*** | right | 21.51 | 22.12 |
| **Model II+** | right + left | 21.86 | 22.40 |

Table 3: Performance comparison of model variants. Models marked with * are variants that employ only the right handshape head. Please note that the experimental setup differs from the **HS-SLR** model presented in Table 2. Here, we unfreeze all parameters in S3D and exclude the optimization of the cross-entropy loss.

### 5.2.2 Pretraining for gloss encoder

We delve into optimal pretraining strategies for the S3D encoder that's coupled with a gloss head. We conduct experiments using **Model I**, as shown in Table 4. We contrast the efficacy of four distinct pretraining methodologies: (1) pretraining solely on Kinetics-400; (2) sequential pretraining, first on Kinetics-400, followed by WLASL; (3) tripletiered pretraining on Kinetics-400, then WLASL, and finally on handshape prediction by attaching two handshape heads while deactivating the gloss head; and (4) a similar three-stage process, but focusing on gloss prediction in the final step.

| Pretrained tasks | Dev | Test |
|---|---|---|
| **Kinetics** | 23.65 | 24.77 |
| **Kinetics + WLASL** | 22.52 | **23.43** |
| **Kinetics + WLASL + HS** | 22.42 | 23.71 |
| **Kinetics + WLASL + Gloss** | **22.31** | 23.71 |

Table 4: Performance of **Model I** with different pertaining strategies. **HS** pretrains the model on predicting the handshapes only (the gloss head is deactivated). **Gloss** pretrains the model on predicting the glosses only (the handshape heads are deactivated).

### 5.2.3 Pretraining for handshape encoder

Table 5 outlines various pretraining strategies adopted for the handshape encoder in **Model II**. The results pertain to right handshape predictions on the PHOENIX14T-HS dataset. Both **Kinetics** and **WLASL** are employed for gloss predictions, as they lack handshape annotations. We also test the **1-Million-Hands** dataset (Koller et al., 2016) for pretraining purposes. This dataset comprises a million images cropped from sign videos, each

| Pretrained tasks | Input | Dev | Test |
|---|---|---|---|
| **None** | full | 52.81 | 50.75 |
| **None** | hand | 73.38 | 72.01 |
| **Kinetics** | full | 26.92 | 27.81 |
| **Kinetics** | hand | 40.56 | 40.18 |
| **Kinetics+WLASL** | full | **25.76** | **25.74** |
| **Kinetics+WLASL** | hand | 43.45 | 40.05 |
| **1-Million-Hands** | full | 47.59 | 46.82 |
| **1-Million-Hands** | hand | 59.04 | 57.68 |

Table 5: Performance of right handshape predictions on PHOENIX14T-HS with various pretraining strategies for the S3D encoder.

| Pseudo-labels | weight | Dev | Test |
|---|---|---|---|
| **Ensemble** | 0.01 | 20.50 | 21.95 |
| **Ensemble** | 0.05 | **20.26** | **21.79** |
| **Ensemble** | 0.1 | 21.22 | 21.77 |
| **Ensemble** | 0.5 | 20.31 | 22.07 |
| **Ensemble** | 1 | 20.63 | 21.86 |
| **HS** | 1 | 21.38 | 22.05 |
| **Ensemble, HS** | 1, 1 | 20.87 | 21.86 |

Table 7: Performance of **Models II** with varying weights and methods for acquiring pseudo-labels used in cross-entropy loss calculation.

labeled with handshapes. To adapt these images for S3D, we duplicate each image 16 times, creating a 'static' video. Furthermore, we experiment with two input formats: the full frame and the right-hand clip. As indicated in Table 5, both pretraining and full-frame input significantly outperform their counterparts.

### 5.2.4 Frozen parameters

We evaluate the impact of freezing varying numbers of blocks within the pretrained S3D encoders in **Model II**. The results are presented in Table 6.

| Frozen blocks | Dev | Test |
|---|---|---|
| **1** | 20.55 | 22.26 |
| **1, 2** | 20.55 | 22.26 |
| **1, 2, 3** | **20.34** | 22.21 |
| **1, 2, 3, 4** | 20.66 | **21.74** |

Table 6: Performance of **Model II** with various frozen blocks in both S3D encoders.

### 5.2.5 Cross-entropy loss

In Table 7, we investigate the computation of cross-entropy loss on handshape predictions utilizing pseudo-labels obtained via two methods: **Ensemble** and **HS**. The former pertains to pseudo-labels gathered as outlined in Section 4.3, while the latter relies on CTC decoder-applied handshape probabilities from the handshape head to produce pseudo-labels. We also examine a hybrid approach (**Ensemble, HS**), which sums the losses from both methods. In addition, we tune the weights $\lambda_{CE}^{L}$ and $\lambda_{CE}^{R}$ in Equation 3, setting them to the same value.

## 6 Conclusions

In this work, we introduce the concept of handshape-aware SLR, enriching this area of re-

search by offering a handshape-enriched dataset, PHOENIX14T-HS, and proposing two distinctive handshape-inclusive SLR methods. Out methodologies maintain orthogonality with existing SLR architectures, delivering top performance among single-modality SLR models. Our goal is to draw increased attention from the research community toward the integration of sign language's phonological features within SLR systems. Furthermore, we invite researchers and practitioners in NLP to contribute to the relatively nascent and challenging research area of SLP, thus fostering a richer understanding from linguistic and language modeling perspectives.

In future work, we would like to explore three primary avenues that show promise for further exploration: (1) Extension of multimodal SLR models. This involves expanding multi-modality SLR models, which use inputs of various modalities like RGB, human body key points, and optical flow, to become handshape-aware. This approach holds potential as different streams capture distinct aspects of sign videos, supplying the system with a richer information set. (2) Contrastive learning. Rather than using handshape labels as supervision, they can be employed to generate negative examples for contrastive learning. This can be achieved by acquiring the gloss segmentation from the CTC decoder and replacing the sign in the positive examples with its counterpart in the handshape minimal pair. The resulting negative examples would be particularly challenging for the model to distinguish, thereby aiding in the development of better representations. (3) Data augmentation. Alternatively, to create negative examples for contrastive learning, the data volume could be increased using the same method that generates negative examples for contrastive learning.

## Limitations

**Noisy labels:** As highlighted in Section 3, the handshape labels we create might be noisy, since two-thirds of them are from a user-edited online dictionary. Additionally, these labels may not correspond perfectly to the sign videos due to the variations among signers and specific signs.

**Single annotator:** Finding DGS experts to serve as our annotators proved challenging. Also, obtaining multiple annotators and achieving inter-annotator agreements proved to be difficult.

**Single parameter:** The dataset used in our study does not account for other sign language parameters including orientation, location, movement, and facial expressions. Moreover, these parameters are not explicitly incorporated as subsidiary tasks within our SLR methodologies.

**Single dataset:** We only extend a single dataset with handshape labels. It remains to be seen whether the methods we propose will prove equally effective on other datasets, featuring different sign languages, domains, or sizes.

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
