# OpenReview forum: "Handshape-Aware Sign Language Recognition: Extended Datasets and Exploration of Handshape-Inclusive Methods"
_EMNLP/2023/Conference — EMNLP 2023 Findings_

### Official Review · Reviewer_NMb4 · 2023-07-26

**Typos Grammar Style And Presentation Improvements:** 500 Out -> Our ?
**Soundness:** 4

**Excitement:**

2: Mediocre: This paper makes marginal contributions (vs non-contemporaneous work), so I would rather not see it in the conference.

**Paper Topic And Main Contributions:**

The work introduces two handshapes "aware" model architectures for sign language recognition (SLR) tasks on the PHOENIX dataset.

**Questions For The Authors:**

A. The best model is the Model II variant, where there are two heads for left and right hands. However, Table 3 indicates low gains from left head (22.07 vs 22.56). This is consistent with the statement that right-handed signers were in the dataset (line 271). Why not remove the left head entirely to save space and improve the training speed?

B. Please specify who and how you could benefit from your paper. (For example: Anyone who is using PHOENIX dataset and works on an SLR task)

**Reasons To Accept:**

This work is well written, and the reporting on the study is sufficient to support its claims.
Contributions of the work are relevant to the ongoing research competition in SLR on the PHOENIX dataset.
The authors claim to enrich the PHOENIX sign language dataset with handshape annotation.

**Reasons To Reject:**

The handshape predictors are valuable contributions. However, the main stated focus is on the SLR task, where the overall performance gain is not substantial (21.8 vs. sota 22.4). That limits the perceived contribution of the work.

It is implemented for well-researched datasets, and the authors did not show how other sign language datasets can directly benefit from their work. That limits the perceived contribution of the work.

The handshape "awareness" of the proposed models does not substantially increase the performance compared to its own vanilla "unaware" approach (22.07 vs. 23.69). However, the increase in memory and training speed is substantial. That diminishes the main contribution claim.

**Reproducibility:**

2: Would be hard pressed to reproduce the results. The contribution depends on data that are simply not available outside the author's institution or consortium; not enough details are provided.

**Reviewer Confidence:**

3: Pretty sure, but there's a chance I missed something. Although I have a good feel for this area in general, I did not carefully check the paper's details, e.g., the math, experimental design, or novelty.

---

> ### Author Rebuttal · Authors · 2023-08-28
>
> Thank you for your review!
> # In response to your questions:
>
> > A.*The best model is the Model II variant, where there are two heads for left and right hands. However, Table 3 indicates low gains from left head (22.07 vs 22.56). This is consistent with the statement that right-handed signers were in the dataset (line 271). Why not remove the left head entirely to save space and improve the training speed?*
>
> Thank you for your astute observation and suggestion. While the gains from adding an extra left-hand head appear to be minimal, as indicated by the comparison of 22.07 vs. 22.56 in Table 3, our intention is to provide a comprehensive examination of the effects of incorporating handshape information into SLR systems. For this reason, we have chosen to report the results for both cases, thereby allowing readers to make an informed decision on whether to include the left-hand head in their own SLR system development.
>
> >B.*Please specify who and how you could benefit from your paper. (For example: Anyone who is using PHOENIX dataset and works on an SLR task)*
>
> Our paper is designed to benefit a broad audience, including both users of the PHOENIX dataset and researchers engaged in SLR tasks.
>
> For those utilizing the PHOENIX dataset, our handshape labels provide a valuable resource for deeper analysis and experimentation. Specifically, our labels enable users to conduct linguistic analyses, gather statistics on handshape distribution, and assess the dataset's suitability for real-world applications. Additionally, users can perform error analyses to identify patterns in the misrecognition of signs with certain handshapes and use the label information to develop more advanced systems. For instance, the labels can facilitate contrastive learning by enabling the creation of negative examples using signs with similar handshapes.
>
> For SLR practitioners, our paper serves as a comprehensive reference and guide for creating new datasets and developing SLR systems for real-world applications. By demonstrating the benefits of handshape-aware SLR, we hope to encourage practitioners to consider handshape labeling as a valuable component of their work. Ultimately, we believe that the insights and strategies presented in our paper will be valuable takeaways for practitioners seeking to advance the field of SLR.
>
> # In response to the critiques:
> > *The handshape predictors are valuable contributions. However, the main stated focus is on the SLR task, where the overall performance gain is not substantial (21.8 vs. sota 22.4). That limits the perceived contribution of the work.*
>
> We appreciate the feedback and understand that the improvement from 22.4 to 21.8 may not seem substantial at first glance. However, it is important to consider this advancement in the context of recent developments in the field of SLR. As indicated in Table 2 of our paper, the state-of-the-art (SOTA) performance has been incrementally improving over time. For instance, MMTLB (Chen et al., NeuRIPS, 2022a) improved the SOTA from 23.9 (SignBT, Zhou et al., CVPR 2021a) to 22.5, and SignBT improved the previous SOTA (CMA, Papastratis et al., IEEE 2020) from 24.0 to 23.9.
>
> While these improvements may seem incremental, they represent important progress in a field where gains are often hard-fought and incremental. Our work contributes to this ongoing effort to push the boundaries of what is currently achievable in SLR. Additionally, our paper introduces the concept of handshape awareness, which we believe is a valuable contribution that opens up new avenues for exploration and improvement in SLR systems.
>
> # In response to the concerns of the reproducibility:
>
> Ensuring reproducibility is of utmost importance to us, and we have taken several steps to facilitate this. As stated in the abstract, we have committed to making our dataset and code publicly available. Additionally, we have made a concerted effort to provide detailed information about the system configuration and experimental setups in the paper.
>
> We understand that reproducibility encompasses various aspects, and we are open to making any additional modifications that may be necessary. If there are specific elements that you feel are missing or could be further elaborated upon to enhance reproducibility, we would greatly appreciate your guidance. We are fully committed to addressing any gaps and providing all the necessary information to ensure that our work can be accurately replicated by others.
>
> We hope that our responses have sufficiently addressed the concerns you raised. Please take our explanations into consideration and reevaluate our paper. We appreciate the time and effort you have invested in reviewing our work. We firmly believe that this research has the potential to make a meaningful impact on the SLR community, and we are grateful for your consideration.

---

### Official Review · Reviewer_TyCu · 2023-08-05

**Soundness:** 3

**Excitement:**

4: Strong: This paper deepens the understanding of some phenomenon or lowers the barriers to an existing research direction.

**Paper Topic And Main Contributions:**

This paper addresses the problem of handshape-aware Sign Language Recognition (SLR) and makes several key contributions in this area.

Handshape-Enriched Dataset: The paper introduces a new dataset called PHOENIX14T-HS, which is an extension of the PHOENIX14T dataset. PHOENIX14T-HS includes handshape labels derived from the SignWriting dictionary and manual labeling. The dataset features German Sign Language (DGS) videos with annotations for glosses and translations into spoken German, along with the newly added handshape labels for both hands.

Handshape-Inclusive SLR Methods: The paper proposes two handshape-inclusive SLR methods, referred to as Model I and Model II. These methods explicitly utilize handshape information and are designed to predict both sign glosses and handshapes concurrently. Model II, in particular, utilizes dual encoders, one for glosses and one for handshapes, and employs a joint head for generating gloss predictions. This approach outperforms existing single-modality SLR models, ranking as the top performer among them.

The main focus of this paper is to promote the integration of handshape information within SLR systems, enrich the existing SLR datasets with handshape labels, and propose effective handshape-inclusive SLR methods. The contributions aim to advance the understanding and development of SLR models that can take into account the phonological features of sign languages.

**Questions For The Authors:**

1. Considering the challenges in finding DGS experts as annotators, how did you address potential biases or inconsistencies in the annotations? Did you perform any inter-annotator agreement analysis to assess the reliability of the handshape labels?

2. While the PHOENIX14T-HS dataset focuses on handshape labels, sign language recognition involves multiple parameters like orientation, location, movement, and facial expressions. How do you plan to address the exclusion of these other important parameters in future work, and how might it affect the performance of the proposed methodologies in more comprehensive sign language recognition tasks?

3. The paper discusses the potential extension of multimodal SLR models to become handshape-aware. Can you elaborate on the advantages and challenges of integrating other modalities beyond handshapes, and how this integration could impact the overall performance and robustness of the proposed approaches?

4. In the evaluation, the paper primarily focuses on accuracy improvements in SLR. Have you considered conducting experiments to evaluate the real-world impact and practical usability of the proposed methods in sign language recognition systems? If not, do you have any plans for such evaluations in future work?

5. The paper emphasizes the performance on the PHOENIX14T-HS dataset. How well do the proposed methodologies generalize to other sign language datasets with different sign languages, domains, and sizes? Have you considered conducting multi-dataset evaluations to assess the generalizability of the methods?

**Reasons To Accept:**

1. Handshape-Enriched Dataset: The paper introduces a new dataset, PHOENIX14T-HS, which includes handshape labels for sign language videos. This dataset represents a valuable resource for researchers and practitioners working on Sign Language Recognition (SLR) and related tasks. It enables the development and evaluation of handshape-aware SLR models, which is crucial for advancing the understanding of sign languages.

2. Advancement in SLR: The proposed handshape-inclusive SLR methods, Model I and Model II, represent significant advancements in the field of SLR. Model II, in particular, achieves top performance among single-modality SLR models. These methods open up new possibilities for exploring multimodal SLR models and further improving SLR accuracy.

3. Pretraining Strategies: The paper thoroughly explores various pretraining strategies for the S3D encoder in SLR models. This analysis provides valuable insights into effective pretraining methodologies for sign language-related datasets, which are often low-resource and challenging to handle. The findings can guide researchers in efficiently utilizing pretraining to boost performance on SLR tasks.

4. Impact on SLR Model Design: By incorporating handshape information into the SLR models, the paper addresses a crucial aspect of sign languages that has been largely underexplored in previous work. This not only improves the accuracy of SLR models but also aligns with the linguistic properties of sign languages, contributing to more linguistically meaningful SLR systems.

5. Pseudo-Label Weighting Analysis: The investigation into pseudo-label weighting for handshape predictions provides useful guidance on how to handle noisy labels and improve the learning process. The results shed light on the effective utilization of pseudo-labels for fine-grained frame-level supervision in SLR.

6. Potential for Further Research: The paper opens up promising avenues for future research in SLR, such as exploring multimodal SLR models, contrastive learning for negative examples, and data augmentation techniques. These areas hold potential for further advancing the performance and understanding of SLR systems.

**Reasons To Reject:**

1. Noisy Handshape Labels: One of the main weaknesses of the paper is that the handshape labels in the PHOENIX14T-HS dataset are potentially noisy. Two-thirds of the handshape labels are obtained from a user-edited online dictionary, which may not always accurately correspond to the sign videos. This could introduce noise and uncertainty in the dataset, affecting the reliability of the results.

2. Single Annotator: The paper mentions that finding experts to serve as annotators for the handshape labels was challenging, and obtaining multiple annotators for inter-annotator agreement was difficult. This limitation raises concerns about the consistency and reliability of the annotations, as inter-annotator agreement is crucial for ensuring high-quality labeled data.

3. Limited Sign Language Parameters: The dataset used in the study only considers handshape labels and does not include other sign language parameters such as orientation, location, movement, and facial expressions. While the focus on handshape is a specific contribution, the exclusion of other important parameters may limit the broader applicability of the proposed methodologies.

4. Single Dataset Evaluation: The paper primarily evaluates the proposed methods on the PHOENIX14T-HS dataset. While this dataset is enriched with handshape labels, it would be beneficial to see how the methods generalize to other sign language datasets, representing different sign languages, domains, and sizes. The lack of multi-dataset evaluation raises questions about the robustness and generalizability of the proposed approaches.

5. Lack of Multimodal Integration: While the paper discusses the potential extension of multimodal SLR models to become handshape-aware, it does not explore the integration of other modalities beyond handshapes. Considering that SLR can benefit from multiple modalities like RGB videos, human body key points, and optical flow, the exclusion of multimodal integration may limit the overall effectiveness of the proposed methodologies.

6. Absence of Real-World Impact Evaluation: The paper primarily focuses on the technical contributions and improvements in SLR accuracy. However, there is limited discussion about the real-world impact of the proposed methods in practical sign language recognition systems. Without a comprehensive evaluation of the practical usability and efficiency of the proposed approaches, it remains uncertain how these methods would perform in real-world applications.

**Reproducibility:**

3: Could reproduce the results with some difficulty. The settings of parameters are underspecified or subjectively determined; the training/evaluation data are not widely available.

**Reviewer Confidence:**

3: Pretty sure, but there's a chance I missed something. Although I have a good feel for this area in general, I did not carefully check the paper's details, e.g., the math, experimental design, or novelty.

---

> ### Author Rebuttal · Authors · 2023-08-28
>
> Thank you very much for your detailed reviews!
> # In response to the questions:
> > 1. *Considering the challenges in finding DGS experts as annotators, how did you address potential biases or inconsistencies in the annotations? Did you perform any inter-annotator agreement analysis to assess the reliability of the handshape labels?*
>
> We recognize the importance of conducting inter-annotator agreement analysis to validate and enhance the quality of handshape labels. However, due to the limited availability of a single annotator, we were unable to conduct such an analysis, which likely resulted in noisy labels. Nonetheless, we would like to emphasize that these noisy handshape labels actually mirror real-world scenarios: handshape labels are inherently noisy due to the ambiguity in determining handshape during coarticulations (the transition between two signs) and the application of an existing sign dictionary. Therefore, while we unintentionally created noisy labels, we have demonstrated that handshape-aware SLR shows promise even in the presence of such noise.
>
> > 2. *While the PHOENIX14T-HS dataset focuses on handshape labels, sign language recognition involves multiple parameters like orientation, location, movement, and facial expressions. How do you plan to address the exclusion of these other important parameters in future work, and how might it affect the performance of the proposed methodologies in more comprehensive sign language recognition tasks?*
>
> Our approach is designed to be easily extendable to incorporate other sign parameters. For example, one could add an extra head to predict, for example, orientation (as in Model I in the paper), or add an extra encoder for orientation (as in Model II in the paper) on top of our existing system. Since this would involve providing more information to the model, we anticipate that it would result in improved performance. However, it should be noted that the weights of the loss from different parameters might be an important hyperparameter to tune carefully.
>
> >3. *The paper discusses the potential extension of multimodal SLR models to become handshape-aware. Can you elaborate on the advantages and challenges of integrating other modalities beyond handshapes, and how this integration could impact the overall performance and robustness of the proposed approaches?*
>
> Our approach is single-modal in that the system inputs are in RGB. However, the system is designed to be readily extendable to a multimodal approach, which would involve adding extra encoders for keypoints and/or optical flow. Keypoints and optical flow capture different aspects of the sign video compared to RGB features. Specifically, the signers’ appearance and background are omitted in the former, while some important details might also be omitted due to the application of an external keypoint or optical flow extractor. By making the handshape-aware system multimodal, we anticipate that the performance and robustness of the system would be further improved. However, it also presents challenges such as the need for more computational resources.
>
> >4. *In the evaluation, the paper primarily focuses on accuracy improvements in SLR. Have you considered conducting experiments to evaluate the real-world impact and practical usability of the proposed methods in sign language recognition systems? If not, do you have any plans for such evaluations in future work?*
>
> Thank you for raising this important question. The primary focus of this paper is to serve as a reference and guide for future work in real-world SLR applications. The PHOENIX dataset, while valuable, has only 7,096 training examples and is concentrated in a specific domain (weather forecast), limiting its applicability to broader real-world scenarios. To facilitate real-world applications and evaluations, there is a pressing need for much larger annotated datasets, representing one of the most urgent challenges in the field of SLR. However, we believe our paper holds significant implications for future developments in SLR. It underscores the importance of annotating handshapes when creating new datasets and demonstrates the potential of building handshape-aware SLR systems. We are eager to test our approach on a comprehensive real-world sign language dataset as soon as one becomes available.
>
> >5. *The paper emphasizes the performance on the PHOENIX14T-HS dataset. How well do the proposed methodologies generalize to other sign language datasets with different sign languages, domains, and sizes? Have you considered conducting multi-dataset evaluations to assess the generalizability of the methods?*
>
> We appreciate your concern regarding the generalizability of our methodologies across different sign languages, domains, and dataset sizes. We fully acknowledged the limitation imposed by validating our results on a single dataset, and concur that testing on additional datasets would afford a more robust justification of our claims. However, it is important to note that the SLR community currently has access to only two readily available sentence-level datasets: the PHOENIX dataset and a Chinese sign language dataset, CSL-Daily. We had initially planned to conduct experiments with CSL-Daily as well, but unfortunately, the SignWriting dictionary has extremely limited coverage of Chinese sign language. This implies that to test on CSL-Daily, we would be required to label the dataset ourselves, a task that, given our limited resources, is nearly unfeasible. We have striven to conduct meaningful research within these constraints.
>
> We are truly grateful for the valuable time you have invested in reviewing this paper and the insightful questions you have posed. We sincerely hope our responses have addressed your questions thoroughly.

---

### Official Review · Reviewer_uuuR · 2023-08-10

**Soundness:** 3

**Excitement:**

3: Ambivalent: It has merits (e.g., it reports state-of-the-art results, the idea is nice), but there are key weaknesses (e.g., it describes incremental work), and it can significantly benefit from another round of revision. However, I won't object to accepting it if my co-reviewers champion it.

**Paper Topic And Main Contributions:**

The paper primarily delves into the concept of handshape-awareness in Sign Language Recognition (SLR). The authors have augmented the PHOENIX14T dataset by introducing handshape labels, leading to the creation of the new PHOENIX14T-HS dataset. Two distinct handshape-inclusive SLR methodologies are proposed: a single-encoder network and a dual-encoder network. A training strategy that concurrently optimizes both the CTC loss and frame-level cross-entropy loss is also introduced. The proposed methodologies consistently surpass the baseline performance.

**Questions For The Authors:**

Question A:
Given the significance of handshapes in SLR, how do you envision the applicability and adaptability of your proposed methods to sign language translation tasks?

Question B:
The paper mentions the potential noise in handshape labels, primarily sourced from a user-edited online dictionary. Could you provide more insights into the validation process of these labels? How do you ensure the reliability of such labels, especially when considering the variations among signers and specific signs?

**Reasons To Accept:**

1. Innovation: The paper introduces the concept of handshape-aware SLR, marking a relatively novel direction in the field.
2. Dataset Contribution: The provision of the handshape-enriched dataset, PHOENIX14T-HS, offers a valuable resource for researchers.
3. Methodology: Two unique handshape-inclusive SLR methodologies are proposed, maintaining orthogonality with existing SLR architectures, and suggesting practical applicability.
4. Future Directions: The paper outlines several promising avenues for future research, offering insights for further exploration by researchers.

**Reasons To Reject:**

1. Insufficient Details and Comparative Analysis:

The paper introduces two distinct SLR methods but could benefit from a more detailed exposition of experimental procedures and validations to firmly back its primary claims.
A deeper comparative analysis with existing works is essential. While other methods utilize multiple modalities beyond just RGB videos, such as human body key points and optical flow, the proposed method in this paper leverages additional labels and is pretrained. This distinction necessitates a more thorough comparison to truly ascertain the advantages of the introduced methods over existing ones.

2. Experimental Validation and Generalizability:
The experimental section might need further elaboration and validation to solidify the paper's primary arguments.
The results are validated on a single dataset, raising concerns about the generalizability of the proposed methods. The potential impact of noisy labels, especially when sourced from a user-edited online dictionary, and the challenges tied to relying on a single annotator further compound these concerns.

**Reproducibility:**

2: Would be hard pressed to reproduce the results. The contribution depends on data that are simply not available outside the author's institution or consortium; not enough details are provided.

**Reviewer Confidence:**

2: Willing to defend my evaluation, but it is fairly likely that I missed some details, didn't understand some central points, or can't be sure about the novelty of the work.

---

> ### Author Rebuttal · Authors · 2023-08-28
>
> Thank you for your review!
> # In response to your questions:
> > *Question A: Given the significance of handshapes in SLR, how do you envision the applicability and adaptability of your proposed methods to sign language translation tasks?*
>
> Thank you for raising this important question. The applicability and adaptability of our proposed methods to sign language translation tasks are indeed promising. Our SLR system can be readily adapted to a sign language translation system, and this adaptability is not unique to our model; it is a feature of most SLR systems. There are two primary approaches to this adaptation: the end-to-end system and the cascaded system, both of which have parallels in speech translation.
>
> For the end-to-end system, we would retain both the gloss and handshape encoders and attach a sequence-to-sequence decoder, such as a transformer decoder, to the system. This would enable videos to be mapped directly to translations in spoken languages. On the other hand, for the cascaded system, we would first obtain the output glosses from the SLR system and then use them as input to train a gloss-to-text machine translation system, which could be a transformer model.
>
> Both approaches have their merits and can be chosen based on specific requirements and constraints of the task.
>
> > *Question B: The paper mentions the potential noise in handshape labels, primarily sourced from a user-edited online dictionary. Could you provide more insights into the validation process of these labels? How do you ensure the reliability of such labels, especially when considering the variations among signers and specific signs?*
>
> We acknowledge that obtaining clean and trustworthy handshape labels is ideal but also inherently challenging due to the application of a dictionary to a sentence-level dataset, and the variations among signers and specific signs. This means that it is not guaranteed that the handshape label from the dictionary will be accurate for signs in the sentence-level dataset. Verifying the accuracy of handshape labels for each example in the dataset is a time-consuming process. Given our limited resources, we simplified the validation process by randomly sampling approximately 100 sentences from the labeled dataset and correcting handshape labels where mistakes were identified.
>
> Despite these challenges, we believe that our approach of applying a dictionary to sentence-level datasets has practical value. The field of sign language recognition is still emerging, and there are limited datasets available. Building larger, real-world reflective datasets is a crucial challenge to address for the community. Our paper underscores the importance of including handshape labels in datasets. However, annotating handshapes and ensuring the accuracy of every handshape in a specific dataset entails a significant additional workload, which would need to be repeated for each new dataset created in the future. A more practical approach is to build a sign dictionary or leverage an existing online dictionary, which could substantially save effort and resources. Although this method comes with unavoidable noise, our research demonstrates that it still leads to improvements in the SLR system.
>
> In conclusion, the inevitability of noise in the labels mirrors real-world scenarios, and our study presents a promising and successful example of leveraging such noisy labels to enhance SLR systems.
>
> # In response to the critiques:
> > 1. *Insufficient details and comparative analysis: while other methods utilize multiple modalities, the proposed method in this paper leverages additional labels and is pretrained.*
>
> We appreciate your feedback regarding the details and comparative analysis of our method. Our study indeed focuses on leveraging additional labels and pertaining.
>
> - Multiple modalities: Our study utilizes a single modality (RGB), consistent with several previous works such as SFL (Niu and Mak, 2020), FCN (Cheng et al., 2020), and others mentioned. As shown in Table 2, our model outperforms these previous works, all of which adopted a single modality. Additionally, our method is adaptable to different modalities; it can be easily extended into a multi-modal system by adding extra encoders, such as keypoint and optical flow encoders, which holds promise for further performance improvement.
>
> - Additional labels: The main focus of our study is to determine whether adding additional labels improves the performance of SLR systems. CNN-LSTM (Koller et al., 2019) is a comparable system that also adopted extra labels, yet our system outperforms theirs (Table 2, 21.8 vs. 24.1 WER).
>
> - Pretraining: Several studies, including SignBT (Zhou et al., 2021a), MMTLB (Chen et al., 2022a), and SMKD (Hao et al., 2021), have adopted pretraining strategies. We not only compare our results with these studies in Table 2 but also provide a thorough examination of various pretraining strategies in section 5.2.2 (Pretraining for gloss encoder) and section 5.2.3 (Pretraining for handshape encoder).
>
> We strive to make our experiments and ablation studies as comprehensive as possible. If there are any specific experiments or details you believe are necessary to better justify our claims, we would be grateful for your suggestions. Our primary claim is that handshape-aware SLR is necessary, and we believe our work provides substantial evidence to support the assertion.
>
> >2.*Experimental validation and generalizability: The results are validated on a single dataset. The potential impact of noisy labels compounds these concerns.*
>
> - Single dataset: We acknowledge the limitation of validating our results on a single dataset and agree that testing on additional datasets would provide a more robust justification for our claims. However, we would like to point out that the SLR community currently has only two readily available sentence-level datasets: the PHOENIX dataset and a Chinese sign language dataset, CSL-Daily. We initially planned to conduct experiments with CSL-Daily as well, but the SignWriting dictionary has extremely limited coverage of Chinese sign language. This means that to test on CSL-Daily, we would need to label the dataset ourselves, a task that, given our limited resources, is nearly impossible. We have done our best to conduct meaningful research within these constraints.
>
> - Noisy labels: We openly acknowledge that our labels are noisy. As mentioned in our response to your second question, we believe that noisy labels are a realistic reflection of real-world scenarios. Moreover, even with noisy labels, our handshape-aware SLR still achieves SOTA results. This underscores the success of our approach, as it suggests that clean labels could potentially lead to even better performance.
>
> # In response to the concerns of the reproducibility:
>
> Ensuring reproducibility is of utmost importance to us, and we have taken several steps to facilitate this. As stated in the abstract, we have committed to making our dataset and code publicly available. Additionally, we have created a concerted effort to provide detailed information about the system configuration and experimental setups in the paper.
> We understand that reproducibility encompasses various aspects, and we are open to making any additional modifications that may be necessary. If there are specific elements that you feel are missing or could be further elaborated upon to enhance reproducibility, we would greatly appreciate your guidance. We are fully committed to addressing any gaps and providing all the necessary information to ensure that our work can be accurately replicated by others.
>
> We sincerely appreciate the time and effort you have invested in reviewing our paper and considering our detailed response. It is our firm belief that this paper will have a meaningful impact on the sign language processing community, and hopefully, we have done our utmost to communicate this belief clearly through both our response and the original paper. We kindly ask you to take all of this into account and reconsider the value of the work. Thank you very much for your consideration!

---

### Official Review · Reviewer_ZC2W · 2023-08-15

**Soundness:** 3

**Excitement:**

2: Mediocre: This paper makes marginal contributions (vs non-contemporaneous work), so I would rather not see it in the conference.

**Paper Topic And Main Contributions:**

Authors tackles the problem of Sign Language Recognition (SLR), and augment the existing the PHOENIX14T dataset with handshape labels, called PHOENIX14T-HS dataset. Then they propose two model variations with single encoder and two encoders. In experimental results, they show that the proposed approach outperforms baseline models of single modality (i.e., video).

**Questions For The Authors:**

- Why Model II+ performs worse than Model II?
- Between the proposed single and two encoder models, it was not clear why two encoder model performs better.

**Reasons To Accept:**

- Authors build a new augment dataset for SLR with handshape annotations, which may be useful for subsequent research.
- This paper proposed the models for SLR and the best model outperforms the baselines of single modality (i.e., video).


**Reasons To Reject:**

- Authors employ single annotators to build the proposed dataset set and it might have noisy labels.
- The proposed model utilizes extra label information (handshape labels) to outperform the baselines, which may not be fair comparisons.


**Reproducibility:**

3: Could reproduce the results with some difficulty. The settings of parameters are underspecified or subjectively determined; the training/evaluation data are not widely available.

**Reviewer Confidence:**

3: Pretty sure, but there's a chance I missed something. Although I have a good feel for this area in general, I did not carefully check the paper's details, e.g., the math, experimental design, or novelty.

---

> ### Author Rebuttal · Authors · 2023-08-28
>
> Thank you for your review!
> # In response to your questions:
> > *Why Model II+ performs worse than Model II?*
>
> Model II is designed with two encoders: a gloss encoder and a handshape encoder. Model II+ is an extension of Model II that includes two additional handshape heads in the gloss encoder, in addition to the gloss head present in Model II. Although this addition would ostensibly improve the model, the Word Error Rate (WER) for Model II+ was 22.40, slightly higher than the 22.07 WER of Model II.
> We believe this unexpected result stems from the learning focus of the gloss encoder. The primary function of the gloss encoder is to predict glosses, however, the addition of handshape heads in Model II+ may have inadvertently shifted the focus towards handshape predictions instead. As the evaluation of the systems is based on gloss predictions, this shift in focus could potentially have a detrimental effect on performance.
> >*Between the proposed single and two encoder models, it was not clear why two encoder model performs better.*
>
> Initially, it was unclear whether the single-encoder or the two-encoder model would yield superior results, which is why we included both versions in our study and compared their performances. Our intention was to provide a comprehensive reference for others in the field of sign language recognition (SLR)  and to suggest that it might be worthwhile to experiment with both models in practical applications.
> Moreover, we hypothesize that the observed performance difference could be related to the learning focus of the encoders, as mentioned in response to your first question. The two-encoder model allows each encoder to specialize in gloss or handshape prediction, potentially leading to better results overall.
>
> We hope this explanation addresses your questions and provides a clearer understanding of our findings.
>
> # In response to the critiques:
> > *Authors employ single annotators to build the proposed dataset and it might have noisy labels.*
>
> We acknowledge the concern regarding the use of single annotators to build the proposed dataset and the potential for noisy labels as a result. We concur that our labels are noisy and that the quality of the dataset could have been enhanced by employing multiple annotators. Nonetheless, we firmly believe that this new dataset will be a valuable resource for future research in sign language recognition.
> Firstly, the currently available PHOENIX14T dataset is relatively small, containing only 7,096 sentences, which limits its utility in real-world applications. Therefore, there is a pressing need to create larger, more comprehensive datasets. Our work, which successfully demonstrates handshape-aware SLR on the PHOENIX14T dataset, underscores the importance of including handshape annotations when developing new datasets.
>
> Secondly, handshape labeling is both costly and inherently noisy due to the challenges associated with determining handshapes during coarticulations (transitions between two signs). However, our study demonstrates that even with handshape labels obtained from a user-edited, publicly available dictionary and a single annotator, it is possible to improve the performance of SLR systems. This is a promising finding, as it suggests that handshape labels are beneficial, even when they are somewhat noisy.
> In conclusion, despite the limitations associated with using a single annotator and potentially noisy labels, our work highlights the value of handshape annotations and provides a foundation for the development of more advanced and practical SLR systems.
> >*The proposed model utilizes extra label information (handshape labels) to outperform the baselines, which may not be fair comparisons.*
>
> The primary objective of our study is to investigate whether the inclusion of additional handshape label information can lead to the development of more effective SLR systems. As such, our intention is not to create an unfair advantage over baseline models but rather to explore ways to enhance the performance of SLR systems in a practical context.
>
> We hope our responses have addressed your concerns adequately. With these considerations in mind, we respectfully request that you reconsider your evaluation of our paper. We are committed to addressing any further questions or concerns you may have and are always open to ongoing dialogue. Thank you for your time!

---

### Meta-Review · Area_Chair_xgbv · 2023-09-27

**Recommendation:** 3

**Metareview:**

This is an interesting paper and will be useful in the relatively new Sign Language Understanding subfield. The new dataset the authors propose, especially integrating a new style of modality in Sign Language (handshapes) would be impactful for multimodal tasks other than Sign Language Understanding as well.

Pros:
- New innovative dataset annotated and publicly available as part of this work with good potential for future work
- Interesting step forward for Sign Language Recognition research

Cons:
- Dataset might have noisy labels due to single annotator
- Lack of Multimodal integration

---

### Decision · Program_Chairs · 2023-10-07

**Decision:**

Accept-Findings

**Comment:**

This is an interesting paper and will be useful in the relatively new Sign Language Understanding subfield. The new dataset the authors propose, especially integrating a new style of modality in Sign Language (handshapes) would be impactful for multimodal tasks other than Sign Language Understanding as well.

Pros:
- New innovative dataset annotated and publicly available as part of this work with good potential for future work
- Interesting step forward for Sign Language Recognition research

Cons:
- Dataset might have noisy labels due to single annotator
- Lack of Multimodal integration